

# Microbiome analysis of *Spodoptera frugiperda* (Lepidoptera, Noctuidae) larvae exposed to *Bacillus thuringiensis* (Bt) endotoxins

Yuliana Castañeda-Molina[1], Sandra María Marulanda-Moreno[1], Clara Saldamando-Benjumea[2], Howard Junca[3], Claudia Ximena Moreno-Herrera[1] and Gloria Cadavid-Restrepo[1]

[1] Departamento de Biociencias/Grupo de investigación Microbiodiversidad y Bioprospección/Laboratorio de Biología Celular y Molecular, Universidad Nacional de Colombia, Medellín, Antioquia, Colombia
[2] Departamento de Biociencias/Grupo de Biotecnologia Vegetal UNALMED-CIB/Laboratorio de Ecología y Evolución de Insectos, Universidad Nacional de Colombia, Medellin, Antioquia, Colombia
[3] RG Microbial Ecology: Metabolism, Genomics & Evolution, Div. Ecogenomics & Holobionts, Microbiomas Foundation, Chía, Cundinamarca, Colombia

Corresponding authors
Yuliana Castañeda-Molina,
yulpcastanedamol@unal.edu.co
Gloria Cadavid-Restrepo,
gecadavi@unal.edu.co

## ABSTRACT

**Background**. *Spodoptera frugiperda* (or fall armyworm, FAW) is a polyphagous pest native to Western Hemisphere and recently discovered in the Eastern Hemisphere. In Colombia, *S. frugiperda* is recognized as a pest of economic importance in corn. The species has genetically differentiated into two host populations named "corn" and "rice" strains. In 2012, a study made in central Colombia demonstrated that the corn strain is less susceptible to *Bacillus thuringiensis* (Bt) endotoxins (Cry1Ac and Cry 1Ab) than the rice strain. In this country, Bt transgenic corn has been extensively produced over the last 15 years. Since gut microbiota plays a role in the physiology and immunity of insects, and has been implicated in promoting the insecticidal activity of Bt, in this study an analysis of the interaction between Bt endotoxins and FAW gut microbiota was made. Also, the detection of endosymbionts was performed here, as they might have important implications in the biological control of a pest.
**Methods**. The composition and diversity of microbiomes associated with larval specimens of *S. frugiperda* (corn strain) was investigated in a bioassay based on six treatments in the presence/absence of Bt toxins and antibiotics (Ab) through bacterial isolate analyses and by high throughput sequencing of the bacterial 16S rRNA gene. Additionally, species specific primers were used, to detect endosymbionts from gonads in *S. frugiperda* corn strain.
**Results**. Firmicutes, Proteobacteria and Bacteroidota were the most dominant bacterial phyla found in *S. frugiperda* corn strain. No significant differences in bacteria species diversity and richness among the six treatments were found. Two species of *Enterococcus* spp., *E. mundtii* and *E. casseliflavus* were detected in treatments with Bt and antibiotics, suggesting that they are less susceptible to both of them. Additionally, the endosymbiont *Arsenophonus* was also identified on treatments in presence of Bt and antibiotics. The results obtained here are important since little knowledge exists about the gut microbiota on this pest and its interaction with Bt endotoxins. Previous studies made in Lepidoptera suggest that alteration of gut microbiota can be used to improve the

management of pest populations, demonstrating the relevance of the results obtained in this work.

## INTRODUCTION

*Spodoptera frugiperda* (J.E. Smith) or fall armyworm (FAW) is a tropical insect that is native to Western Hemisphere (*Cano-Calle, Arango-Isaza & Saldamando-Benjumea, 2015*; *Nagoshi et al., 2020*) and can feed over 350 plant species (*Montezano et al., 2018*). In 2016, FAW was reported in West and Central Africa in corn crops (*Goergen et al., 2016*), in India since 2018 (*Ganiger et al., 2018*), and in 19 Asian countries including Myanmar, China, Vietnam, Japan, Korea, and Queensland, Australia since 2020 (*Nayyar et al., 2021*). In Colombia, *S. frugiperda* is recognized as the most important pest of corn (*Zea mays*), and a secondary pest in sorghum (*Sorghum* spp.), cotton (*Gossypium hirsutum*), and pasture grasses (*Cano-Calle, Arango-Isaza & Saldamando-Benjumea, 2015*). This species has diverged into two strains that differ in their genetics and have been named the corn and the rice strains (*Prowell, McMichael & Silvain, 2004*). In Colombia, the corn strain is usually found in corn, cotton, sorghum, and sugar cane, and the rice strain is mainly found in rice and pasture grass (*Cano-Calle, Arango-Isaza & Saldamando-Benjumea, 2015*). Both have been differentiated with several molecular markers including a PCR-RFLP of the mitochondrial gene COI and Sac, a PCR of the nuclear gene FR, and sequencing of the TPI gene (*Nagoshi et al., 2020*). The strains exhibit differences in resistance to chemical and biological controls in Central Colombia. *Ríos-Díez, Siegfried & Saldamando-Benjumea (2012)* demonstrated that the corn strain is less susceptible to *Bacillus thuringiensis* (Bt) endotoxins (Cry1Ac and Cry1Ab) in laboratory conditions than the rice strain and *Ríos-Díez & Saldamando-Benjumea (2011)* found that the rice strain is more resistant to insecticides (lambda-cyhalothrin and methomyl) than the corn strain.

*Bacillus thuringiensis* (Bt) is a soil-borne bacterium that displays toxicity in a wide variety of arthropods through the production of pore-forming proteins or Cry proteins ($\delta$-endotoxins) (*Bel Cortés, Ferré Manzanero & Hernández-Martínez, 2020*; *Palma et al., 2014*) that bind and insert in midgut cells leading to pore formation and osmotic cell shock resulting in insect's death (*Pardo-López, Soberón & Bravo, 2013*). Nowadays, the use of transgenic crops expressing Cry toxins, is a relatively widely used strategy, to control ground pests, while simultaneously reducing the use of conventional insecticides (*Benbrook, 2012*; *Paddock et al., 2021*). However, their extensive use has increased the resistance in non-target pests to Bt, and thus new alternatives are required for their management, amongst them the use of gut microbiota (*Li et al., 2020*; *Li et al., 2021*; *Paddock et al., 2021*). According to the results obtained by *Zenner de Polanía & Álvarez Alcaráz (2008)*, in central Colombia, Bollgard® transgenic crops have been efficient to control species such as *Heliothis virescens*, *Helicoverpa zea* since these two moths were susceptible to the Cry1AC

endotoxins as their larvae died with LC5O concentrations under laboratory conditions. On the contrary, *Spodoptera frugiperda* and *S. sunia* tolerated very high concentrations of this endotoxin (between 192 and 1.178 mg/mL) demonstrating that this type of crop can be more efficiently used for some Lepidopterans but not for others.

Microorganisms play a crucial role during the growth and development of insects and also in their immune response and in levels of susceptibility to insecticides and *Bacillus thuringensis* (Bt) endotoxins (*Almeida et al., 2017*; *Orozco-Flores et al., 2017*; *Li et al., 2020*; *Li et al., 2021*; *Paddock et al., 2021*). Studies made with gut microbiota in *Spodoptera littoralis* have shown that the genus *Enterococcus* is beneficial to the species by protecting the insect host from pathogenic microorganisms' toxins (*Shao et al., 2017*). Also, other authors have found that the genus *Bacillus* helps insects to digest and to absorb nutrients by producing lipases, amylases, and proteases (*Ramya et al., 2016*; *Regode et al., 2016*; *Nguyen et al., 2018*). Gut microbiota in insects can also influence the reproduction of pathogens and the development of insecticide resistance, which has become the focus of many insect gut microbiology studies (*Li et al., 2022*). Several studies have been made with the species *Plutella xyllostela*, for example, *Xia et al. (2018)* found that gut bacteria enhance the insecticidal activity of the Cry toxin protein by causing bacterial septicemia in this moth. Also, *Li et al. (2021)* evidenced that in *P. xylostella* larvae, Bt Cry1Ac protoxin interacts with the gut microbiota by accelerating their mortality. They observed that Cry1Ac protoxin causes a dynamic change in the insect midgut and hemocoel microbiota, with significant increases in both bacterial load and reduction in bacterial diversity. Additionally, *Paddock et al. (2021)* suggested that less susceptibility to Bt produces alterations in the microbiome of *Diabrotica virgifera* (western corn rootworms). They also observed that resistant populations to Bt harbored less rich and distinct communities than susceptible ones.

With respect to endosymbionts and insects of economic importance, such as *Wolbachia*, the transinfection of *Aedes aegypti* with this maternally inherited, endosymbiotic bacterium is a promising new biocontrol approach for the spread of pathogen disease (*Iturbe-Ormaetxe, Walker & O'Neill, 2011*). Also, *Salunkhe et al. (20l4)* detected the presence of *Wolbachia* sp. in *Spodoptera litura* from India by using Multi Locus Sequence Typing (MLST). This bacterium was also found in *Spodoptera exempta* from Tanzania (*Graham et al., 2012*). The genus *Wolbachia* has been differentiated into eleven supergroups (A-K) and in *S. litura* the supergroup B was identified. Nevertheless, *Dumas et al. (2015)* failed to detect *Wolbachia* sp. in *S. frugiperda* based on NGS studies. Recently, *Schlum et al. (2021)* detected short fragments (<100 bp) matching to the RefSeq *Wolbachia* in this pest. However, further studies are required to certify that this endosymbiont is associated to FAW.

This endosymbiont was identified in three species of thrips (*Frankliniella gardeniae*, *F. panamensis* and *Scirtothrips hansoni*) from Colombian avocado where almost 90% of the microbiota of *S. hansoni* was composed of this bacterium according to NGS analyses (*Cano-Calle et al., 2022*). On the other hand, in *Gynaikothrip suzeli* (another thrips species) the endosymbiont *Arsenophonus sp.* was found in almost the entire microbiota of the insect promoting its thelytoky behavior and resulting in the killing of males (*Tyagi et al., 2022*).

Given that studies based on *S. frugiperda* gut microbiota and its interaction with Bt endotoxins have not been made, and also since the detection of endosymbionts are relevant for biological control, the objectives of this work were: (a) to identify the gut bacteria of *S. frugiperda* corn strain in the presence/absence of Bt endotoxins and antibiotics and (b) to detect additional endosymbionts in this insect as no further studies on this subject have been recently made in this pest. In Colombia, transgenic (Bt) corn is extensively produced since 2005 (*Zenner de Polanía & Álvarez Alcaráz, 2008*) and studies based on the response to Bt endotoxins have shown that *S. frugiperda* corn strain is more tolerant to Bt than the rice strain (*Ríos-Díez, Siegfried & Saldamando-Benjumea, 2012*). Gut microbiota manipulation may be a promising method together with other control alternatives to improve the management of this pest since it has rapidly segregated to several countries across the world in corn crops (*Paddock et al., 2021*; *Li et al., 2020*; *Nagoshi et al., 2020*).

## MATERIALS & METHODS

**Ethics statement** The collection of the larvae and genetic access was provided by the ANLA (Autoridad Nacional De Licencias Ambientales to Universidad Nacional de Colombia. Permiso marco de recolección de especímenes silvestres, resolución 0255, 14/03/2014 (artículo 3).

### Larvae collection

*S. frugiperda* larvae (third to the sixth instar, $N = 300$) were sampled from individual corn field plants in two locations are in two different ecoregions 262 km apart, one in the central range of the Northern Colombian Andes at the municipality of Piedras, Tolima, Colombia (4°32′36″N, 74°52′40″W) and at Estación Agraria 'Cotove' farm - Universidad Nacional de Colombia, located in the municipality of Santa Fe de Antioquia, (6°31′54.0″N 75°49′33.8″W) at the western range of the Northern Colombian Andes. The larvae were collected during September 2019.

These larvae were transported to an insect room in the ecology and evolution laboratory, at Universidad Nacional de Colombia (Medellín). They were individually separated into cups of 1.5 oz to avoid cannibalism and were fed following the diet provided by Arevalo and de Polania, (2009), however, formaldehyde was not added, instead of this, was sterilized to avoid contamination. To obtain the F1 generation, the collected larvae were fed on corn leaves, their adults were crossed (crosses were made between adults of Antioquia and Tolima to avoid inbreeding), and their eggs were checked every other day until eclosion. The larvae used for the experiments were fed on a bean diet supplemented with/without antibiotics and with/without Bt. The colony was reared under controlled conditions (28 ± 5 °C and 70% RH) and genotyping of 10 larvae already established from the collections was performed by using the protocol used by *Higuita Palacio et al. (2021)* to identify *S. frugiperda* corn and rice strains on larvae heads by using a PCR-RFLP of the COI gene at the mitochondrial DNA.

### Bt bioassays

Interaction between Bt and gut bacteria was evaluated in 3rth instar larvae of *S. frugiperda* ($N = 180$, 30 larvae per treatment) following Orozco -Flores et al. (2017) protocol and

the following treatments: (1) larvae fed with artificial diet (control), (2) larvae exposed to the artificial diet containing Bt at the LC50, (3) larvae exposed to an artificial diet containing Bt and antibiotics (Bt + Ab), (4) larvae fed on diet with antibiotics for 24 h and subsequently exposed to Bt-treatment in sterile diet (SD) with no antibiotics (Bt + Ab-SD), (5) larvae fed on diet with antibiotics (Ab), and (6) larvae fed on diet with antibiotics for 24 h and subsequently transferred to SD with no antibiotics (Ab-SD). To conduct this bioassay, the product BT-BIOX (constituted by Cry1Aa, Cry1Ab, Cry1Ac, and Cry2A endotoxins) was employed. This commercial product was combined with the bean artificial diet following the *Ríos-Díez, Siegfried & Saldamando-Benjumea (2012)* procedure with a concentration of 1,359.23 ng/ml on the beans diet. Nevertheless, given that *S. frugiperda* larvae showed higher tolerance to the Bt endotoxins in the laboratory, a concentration of 20.0000 UI/cm3 was necessary. No larvae died under this concentration, but fitness was affected (mobility, feeding behavior). Additionally, gut microbiota was eradicated from larvae guts with an antibiotic solution of 5.000 µg/ml (each) of rifampin, gentamicin, tetracycline, streptomycin, and ampicillin placed in a Petri dish with the artificial bean diet. Further on, whole larvae guts were dissected after feeding from each bioassay during 5 days of exposition to extract their DNA for the following analyses.

## Gut dissection

Larvae obtained from each treatment were cooled at −20 °C for 10 min, to reduce vital functions. Later on, they were first washed with ethanol (70%) and then with PBS+Tween20 solution (1%). Gravimetric data were taken from each processed larva. Dissections were performed with sterile forceps in a PBS 1X buffer (phosphate-buffered saline). The entire gastrointestinal tract was extracted, weighed, and macerated in sterile PBS 1X. Each intestinal homogenate was preserved and processed at low temperatures. Half of the homogenate was used for conventional culture-based microbiological methods and the other half was frozen and used for the culture-independent molecular approaches (*Higuita Palacio et al., 2021*).

## Culture dependent analysis
### Bacterial isolation and identification

Gut homogenates from larvae obtained from each treatment ($N = 6$), were cultured by using three larvae pools per treatment ($N = 18$) and then, serial dilutions (up to $10^6$) were plated on a Luria Bertani (LB) medium and were incubated at room temperature for 24 h as indicated by *Higuita Palacio et al., 2021* and then at 37 °C. The rest of the larvae tissue was preserved in ethanol 70% at −20 °C.

After incubation, colony-forming units (CFU) were counted and statistically analyzed with an ANOVA test by using Rstudio (*R Core Team, 2022*). To obtain bacterial cultures, a total of 18 gut samples (from the last instar larvae and average weight = 0.5 g) were used as they corresponded to three replicates per treatment ($N = 6$). Colony bacterial counts were made from Petri dishes with 30 to 300 CFU with dilutions between $10^{-4}$ to $10^{-5}$. Estimation of CFU was made per 1 gr of gut tissue. Colonies were selected according to their morphological differences (size, color, surface, shape, elevation) and isolates were

purified and characterized with Gram staining. Later on, they were conserved in glycerol 20% v/v at −20 °C (*Vivero et al., 2016*).

Pure isolates were grouped by RISA-PCR and selected representatives were identified by using 16S rRNA gene analysis (*Higuita Palacio et al., 2021*). Briefly, DNA from the isolates was extracted and amplified using the primers L1 (5′-CAA GGC ATC CAC CGT-3′) and G1 (5′-GAA GTC GTA ACA AGG-3′), as reported by *Jensen, Webster & Straus (1993)*. PCR products were visualized and a dendrogram with ITS region-banding patterns was constructed by the GelCompar II 6.6 software (Applied Maths NV, Sint-Martens-Latem, Belgium) a cluster analysis was performed using the Pearson coefficient and Simple Linkage cluster method. ≥75% similarity between ITS standards was established as criteria for selecting bacterial strains for subsequent molecular identification assays (*Higuita Palacio et al., 2021*). Total DNA from the selected colonies was used to amplify the 16S rRNA gene, using Eubac 27F (5′-AGA GTT TGA TCC TGGCTC AG-3′), 1492R (5′-GGT TAC CTT GTT ACG ACT T-5′), and the reaction conditions were carried out following *Moreno, Romero & Espejo (2002)* procedure. Some isolates were also amplified by the gene gyrB (gyrase) using Primers UP1 (5′–AGC AGG GTA CGG ATG TGC GAG CCR TCN ACR TCN GCR TCN GTC AT –3′) and UP2R (5′–GAA GTC ATC ATG ACC GTT CTG CAY GCN GGN GGN AAR TTY GA –3′) in order to confirm the identity following *Yamamoto & Harayama (1995)* protocol (*Higuita Palacio et al., 2021*). All PCR products obtained with both molecular markers were visualized in agarose gels 1.2% stained with EZ VisionTM (Amresco, U.S.A).

Amplified products were sequenced on an ABI PRISM 3100 Genetic Analyzer (Applied Biosystems, Carlsbad, CA, USA). The sequences were analyzed with BLAST (Basic Local Alignment Search Tool) and SeqMatch search tool from the Ribosomal Database Project (RDP) (https://rnacentral.org/expert-database/rdp) to identify the gut bacteria associated with *S. frugiperda*. 16S rDNA consensus sequences were edited and aligned using Geneious prime 2021 v 2.2 software and compared with GenBank, RDP (*Cole et al., 2009*), and SILVA database using BLASTN (National Center for Biotechnology Information; http://www.ncbi.nlm.nih.gov/BLAST/) to confirm identity percentage. Alignments were made with ClustalW in Geneious. Bayesian trees were constructed using Mr. Bayes 3.0 (*Huelsenbeck & Ronquist, 2001*). The sequences obtained here can be found in the NCBI GenBank databases with accession numbers OP999646–OP999647, OQ344654–OQ344662 and OQ351359.

## Culture-independent assays
### Microbiome composition by 16S rRNA gene Illumina amplicon sequencing
Total genomic DNA was extracted from homogenized tissue samples using "FastDNA™ SPIN KIT" (MP Biomedicals), following the manufacturer's instructions. DNA concentration was measured and the DNA integrity was analyzed as described previously by *Higuita Palacio et al. (2021)*.

16S rRNA gene amplicons obtained from all treatments were sequenced using the MiSeq Illumina platform. From raw pair-end sequence datasets from each sample (reads of 253 bp), the DADA2 software package (https://github.com/benjjneb/dada2)

was used following a sequential pipeline for filtering, denoising, chimeras, and merging (*Callahan et al., 2016*). Assembled data sets consisted of reads that were trimmed based on sequence quality, the primer sequences were cut off and potentially chimeric sequences were deleted. To detect the counts of each unique Amplicon Sequence Variant (ASV) across all samples was employed to classify them using RDP Naive Bayesian Classifier (*Wang et al., 2007*), using as taxonomic reference the Silva database release 138.1 (https://www.arb-silva.de/documentation/release-138.1/). In all cases, the coverage was above >0.9 with 10.000 reads, therefore Bray-Curtis dissimilarity matrix of non-rarified relative percentages was used. For Bray-Curtis dissimilarity between all pairs of samples at the ASV level was calculated using phyloseq (default parameters, https://joey711.github.io/phyloseq/distance.html). Identical nodes on displayed UPGMA dendrogram clustering were obtained with data rarified to 20.000 reads on each dataset. The phyloseq software package (https://joey711.github.io/phyloseq/) (*McMurdie & Holmes, 2013*) and Microbiome Analyst (https://www.microbiomeanalyst.ca) (*Dhariwal et al., 2017*; *Chong et al., 2020*) were used for estimating and obtaining plots of alpha diversity (within-sample) of Observed, Chao 1, Shannon, Simpson metrics. Beta diversity was also estimated between all communities (Hierarchical clustering analysis, Principal Coordinates Analysis of weighted Unifrac distances-PCoA, Heatmap) and PERMANOVA analysis was finally performed to test for significant differences amongst treatments.

## Endosymbionts detection with conventional PCR

Total genomic DNA was extracted from ($N = 5$) males and ($N = 5$) females' gonads obtained from the laboratory colony. For endosymbionts detection, five individualswere chosen for DNA extraction. The total genome was used for the specific detection of endosymbionts, using the primers: Spiro1 (5′GGAACCTTACCTGGGCTAGAATGTATT3′) and Spiro 2 (5′GCCACTGTCTTCAAGCTCTACCAAC3′) for *Spiroplasma* detection (*Goryacheva et al., 2018*); ArsF (GGGTTGTAAAGTACTTTCAGTCGT), ArsR2 (GTAGCCCTRCTCGTAAGGGCC) for *Arsenophonus* (*Duron et al., 2008*), CLOR1 (5′GGAACCTTACCTGGGCTAGAATGTATT3′) and CLOF1 (5′GCCACTGTCTTCAAG CTCTACCAAC3′) for *Cardinium* (*Duron et al., 2008*); WSP81F (5′TGGTCCAATAAGT GATGAAGAAAC3′), WSP691R (5′AAAAATTAAACGCTACTCCA3′) for *Wolbachia* (*Zhou, Rousset & O'Neill, 1998*; *Braig et al., 1998*); and SS18F (−5′GTTGATTCTG CCTGACGT3′) and SS1492R (5′GGTTACCTTGTTACGACTT3′) (*Ghosh & Weiss, 2009*) for Microsporidia. The PCR conditions for each endosymbiont are given by their authors. PCR products were visualized, amplified, sequenced and analyzed as described above. The sequences obtained here can be found in the NCBI Gene Bank databases with accession numbers OP999648–OP999652.

## RESULTS

### Strain identification

COI amplification of mitochondrial DNA and its subsequent digestion with the *MspI* enzyme made on 10 larvae, allowed us the identification of *S. frugiperda* corn strain from
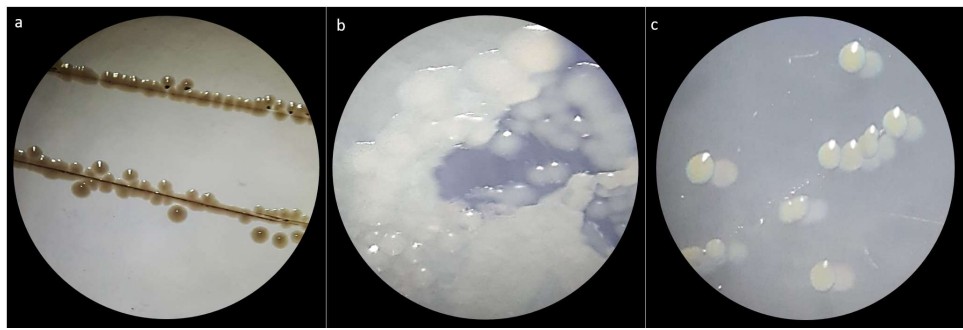

**Figure 1** **Macroscopic colony morphotype.** (A) Morphotype C1. (B) Morphotype C2 and (C) Morphotype C3.

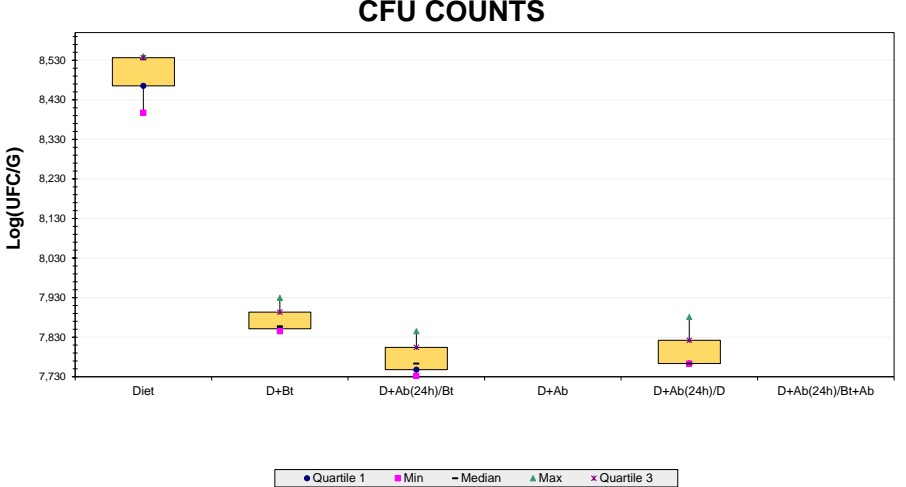

**Figure 2** **CFU counts obtained from Culture-dependent methods.**

the laboratory colony (*Cano-Calle, Arango-Isaza & Saldamando-Benjumea, 2015*; *Higuita Palacio et al., 2021*) (Fig. S1).

## Bacterial diversity through culture-dependent assays

Three bacterial morphotypes were identified on the six treatments and they were labeled C1 (beige and the most abundant; Fig. 1A), C2 (white and intermediately abundant; Fig. 1B), and C3 (white punctiform and less abundant; Fig. 1C). CFU counting on the six treatments were significantly different according to an ANOVA test ($F = 18662.16$, $df = 5$, $p$-value<0.01). Treatments that consisted of diet (the control) grew the majority of CFU and treatments in presence of antibiotics had a drastic CFU reduction (Fig. 2). This result confirms a successful eradication of cultivable gut bacteria from *S. frugiperda* larvae.

**Table 1  Taxonomic identification of *S. frugiperda* gut isolates using 16S rDNA and *GyrB* sequencing.**

| Isolate ID | Origin | NCBI-GenBank Accession number | Phylogenetic affiliation | % Similarity |
|---|---|---|---|---|
| 3C2R1 | Treatment 3 (BT+Ab), morphotype 2 | OQ344661 | *Enterococcus casseliflavus* | 100% |
| 2C3R2 | Treatment 2 (BT LC50), morphotype 3 | OQ344659 | *Enterococcus silesiacus* | 100% |
| 3C1R2 | Treatment 3 (BT+Ab), morphotype 1 | OP999647 | *Enterococcus mundtii* | 95.18% |
| 3C3R2 | Treatment 3 (BT+Ab), morphotype 3 | OQ344660 | *Enterococcus mundtii* | 100% |
| 5C2R1 | Treatment 5 (BT+Ab-SD), morphotype 2 | OP999646 | *Enterococcus gallinarum* | 94.34% |
| SP75 | Resistant morphotype, antibiogram analysis | OQ344658 | *Enterococcus casseliflavus* | 100% |
| SP76 | Resistant morphotype, antibiogram analysis | OQ344654 | *Enterococcus casseliflavus* | 100% |
| 2C1R1 | Treatment 2 (BT LC50), morphotype 1 | OQ344656 | *Enterococcus mundtii* | 100% |
| 1C2R1 | Treatment 1 (Diet), morphotype 2 | OQ344657 | *Enterococcus casseliflavus* | 99.86% |
| 1C2R2 | Treatment 1 (Diet), morphotype 2 | OQ344662 | *Enterococcus casseliflavus* | 100% |
| 5C2R3 | Treatment 5 (BT+Ab-SD), morphotype 2 | OQ344655 | *Enterococcus mundtii* | 99.31% |
| 5C3R1 | Treatment 5 (BT+Ab-SD), morphotype 3 | OQ351359 | *Enterococcus mundtii* | 99.89% |

## Ribosomal intergenic spacer analysis (RISA) of isolates

A total of 36 bacterial isolates were obtained from the cultures. They were further purified and stored at −20 °C. These isolates were analyzed by using the bands of the ITS (Internal transcribed spacer) region between 16S and 23S rDNA. Their band patterns were visualized with Gelcompar II® (Applied Biosystems NV, Sint-Martens-Latem, Belgium). This software differentiated 12 OTUS (clusters) with a similarity percentage of 75%. Representative isolates of each cluster ($N = 12$) were further selected for 16S rRNA gene sequencing and a macro and microscopic characterization. (Fig. S2)

## Identification of isolates with 16S rRNA gene and *gyrB* sequences

Molecular identification with 16S rDNA and *gyrB* showed high percentages of similarity with NCBI sequences databases at the species level (Table 1). These sequences were phylogenetically analyzed by using Bayesian trees for each gene separately (Figs. S3 and S4). Amongst isolates, the species *Enterococcus mundtii*, *E. casseliflavus*, *E. gallinarum*, and *E. silesiacus* were identified.

E. casseliflavus was the most predominant species as it was observed in 6 isolates, followed by *E. mundtii* in five isolates. *E. silesiacus* was only identified on isolate 2C3R2 with a percent identity of 100% to the databases. Also, the three colony morphotypes (C1, C2, and C3) were observed in *E. mundtii*. *E. casseliflavus* was mainly represented by morphotype C2. Isolate 5C2R1 was identified as *E. gallinarum* with a percent identity of 94.34% and corresponded to morphotype C2. Isolate 2C3R2 corresponded to morphotype C3 and was identified as *E. silesiacus* with a percentage of 100% and isolates Sp75 and Sp76, which were resistant to antibiotics, were identified as *E. casseliflavus*.

## Bacterial diversity through high throughput sequencing of the bacterial 16S rRNA gene

After filtering and cleaning the raw data, for low-quality or chimeric sequences, the resulting dataset consisted of 790.011 reads with an average of 79.001 reads per sample ($N = 10$) from which a total of 2.439 ASV were assigned. Reads below 0.0001% of depth were removed from the further analyses obtaining a total of 53 ASV of low quality. Rarefaction results (Fig. S5) showed that the sequence coverage was more than 98% for all six treatments, signifying that the patterns of composition are representative of amplicons complexity and samples can be compared regarding alpha and beta diversity on specimens of *S. frugiperda* corn strain (*Higuita Palacio et al., 2021*).

Results obtained from the six treatments and NGS showed that the phyla Firmicutes, Proteobacteria, Desulfobacterota, Deferribacterota, Campilobacterota, Bacteroidota, and Actinobacteriota were the major groups in *S. frugiperda* (corn strain) gut microbiota. Firmicutes was the most abundant phylum with a relative percentage of 98% in the control (Diet), followed by 96% found in the treatment Diet + Bt (Fig. 3). In diets supplied with antibiotics, the most abundant phylum was also Firmicutes. This phylum considerably reduced its relative abundance to 44% in the treatment Diet + Ab and to 43% in the treatment Diet + Ab + Bt (during 24h).

At the genus level, *Enterococcus* had a relative abundance of 82% in the diet treatment (control), this genus was also the most abundant (77%) in the treatment Diet + Bt. In the assay of larvae treated with Diet + Ab, this genus drastically reduced to 5% (relative abundance), in the treatment of Diet +Ab (24h) to 4%, and in the treatment of Diet +Ab +Bt to 11% (Fig. 4). The most representative genera in all treatments were: *Enterococcus*, *Weisella*, *lleibacterium*, *Burkholderia*, *Ralstonia*, *Dubosiella*, *Bacillus*, *Bifidobacterium*, and *Turicibacter*. In the treatment Diet +Ab (24h) the genus *Weisella* displaced the genus *Enterococcus* (from 61% (Diet treatment) to 15% (Diet +Ab)). Larvae that were fed with antibiotics supplied diets harbored the genera *Ileibacterium* (13%), followed by *Ralstonia* (12%) and *Burkholderia*. Additionally, larvae submitted to diets with Bt, harbored the genera *Burkholderia* (15%) and *Ileibacterium* (14%).

According to the results of the core analysis (Fig. S6), the phyla that remain unchanged in their composition throughout the entire microbial community were: Firmicutes (100% prevalence), Proteobacteria and Actinobacteriota (83% prevalence), and Campilobacterota (50% prevalence). Alpha diversity results based on Chao1 ($F = 1.27$, *p*-value<0.398), Shannon ($F = 5.42$, *p*-value<0.101) and Simpson ($F = 4.014$, *p*-value<0.142) indexes showed no significant differences amongst treatments: Diet, Diet + Ab and Diet + Bt (Fig. 5). However, this diversity was statistically different ($p < 0.0005$) between the treatments without antibiotics (WoAb) *vs.* Diet with antibiotics (at some point in the bioassays either for 24 h or during the 5 days) suggesting that antibiotics eradicated most of the *S. frugiperda* gut microbiota to then allowed a possible microbiota recovery during the rest of the bioassays days. For treatments without antibiotics (WoAb)/Diet with antibiotics (Wab), the estimates were: Chao1 ($F = 2.31$, *p*-value<0.248), Shannon ($F = 4.34$, *p*-value<0.0137),

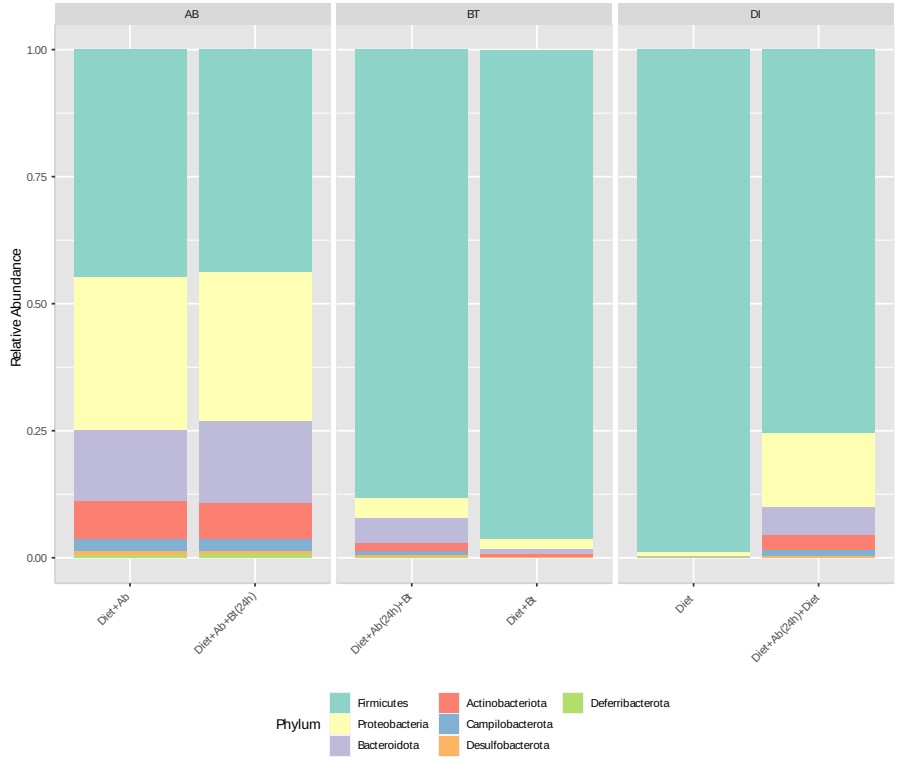

**Figure 3 Relative abundance of phyla observed in *S. frugiperda* gut microbiota analyzed in six treatments.** (1) larvae fed with artificial diet (control), (2) larvae exposed to artificial diet containing *Bt* at the LC $_{50}$, (3) larvae exposed to artificial diet containing *Bt* and antibiotics (*Bt* + Ab), (4) larvae fed on diet with antibiotics for 24 h and subsequently exposed to *Bt*-treatment in sterile diet with no antibiotics (*Bt* + Ab-SD), (5) larvae fed on diet with antibiotics (Ab), and (6) larvae fed on diet with antibiotics for 24 h and subsequently transferred to SD with no antibiotics (Ab-SD).

and Simpson ($F = 5.02$, $p$-value<0.0081) (Fig. 6). It is important to mention that bioassays supplied with antibiotics showed the highest microbiota diversity.

The principal coordinate's analysis (PCoA) (with Bray Curtis dissimilarities) showed two main clusters that explained the 78.6% of beta diversity ($\beta$) in *S. frugiperda* gut communities. The first cluster was composed by samples from assays with antibiotics and the second by assays free of antibiotics ([PERMANOVA] $F$-value =3.0815; R-squared = 0.67259; $p$-value <0.2) (Fig. 7).

A hierarchical cluster analysis together with a heat map and ASV data set showed the presence, at low relative abundance (%) of the genus *Arsenophonus* on the treatments Diet +Ab (24h) + Bt and Diet +Ab (24h) (Fig. 8).

## Endosymbionts detection

Amongst the five endosymbionts analyzed (Table 2), the genus *Arsenophonus* was detected (Fig. 9). BLASTN comparison of the partial 16S rRNA sequence against the NCBI reference database shows 99% similarity to the 16S rRNA gene found in the genome of the type strain *Arsenophonus nasoniae* strain ATCC 49151 NCBI reference (NR_042811.1).

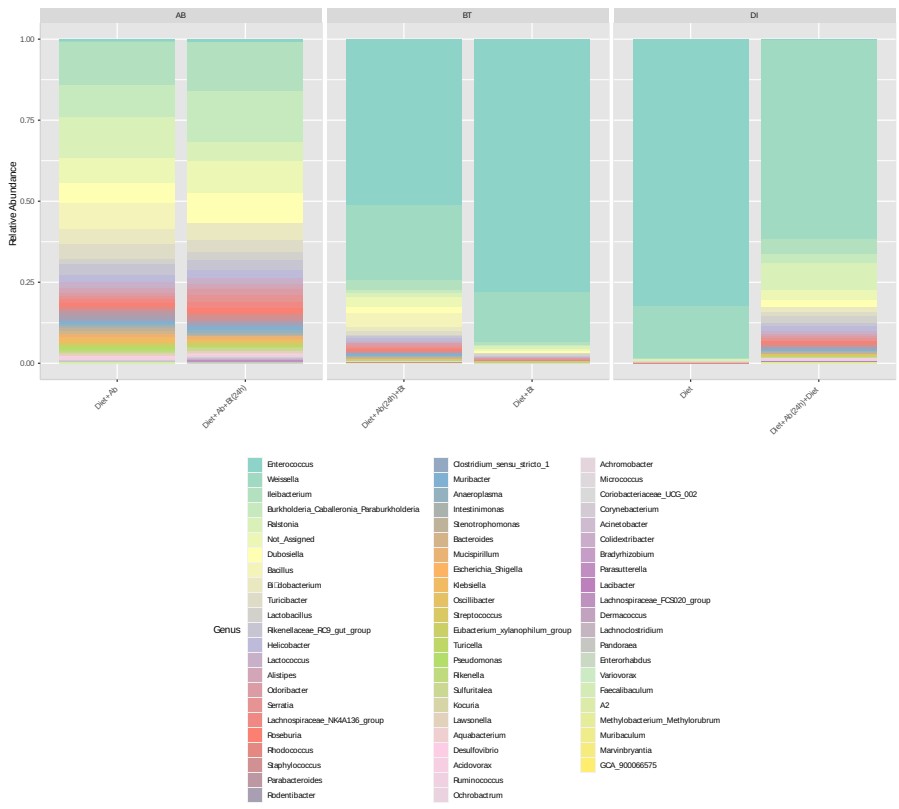

**Figure 4** Relative abundance of genera observed in *S. frugiperda* gut microbiota analyzed in the six treatments.

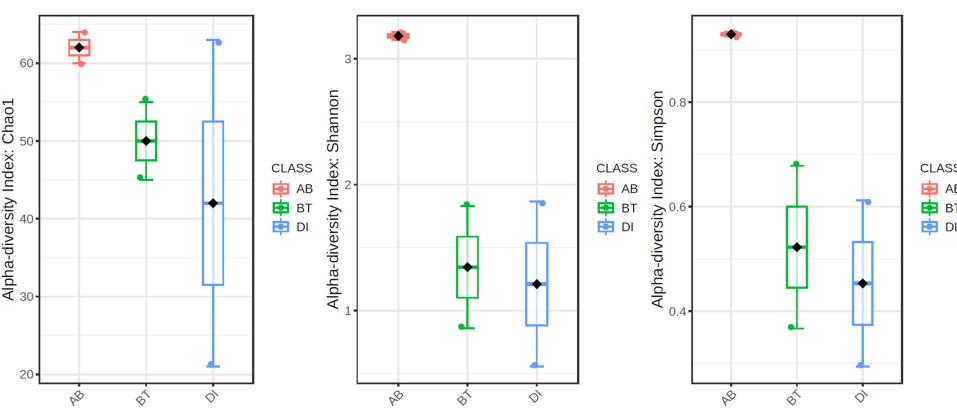

**Figure 5** Comparison amongst treatments AB, BT and DI. AB (assays with antibiotics, Diet+Ab, Diet+Ab+*Bt* (24h)), BT (assays with *B. thuringiensis,* Diet+*Bt,* Diet+Ab (24h)+*Bt*), DI (Diet, Diet+Ab (24h)).

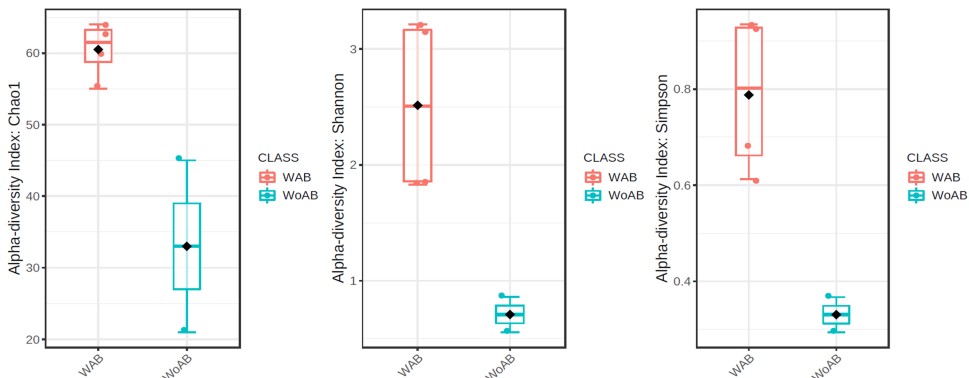

**Figure 6  Comparison between treatments with (WAb)/without antibiotics (WoAb).**

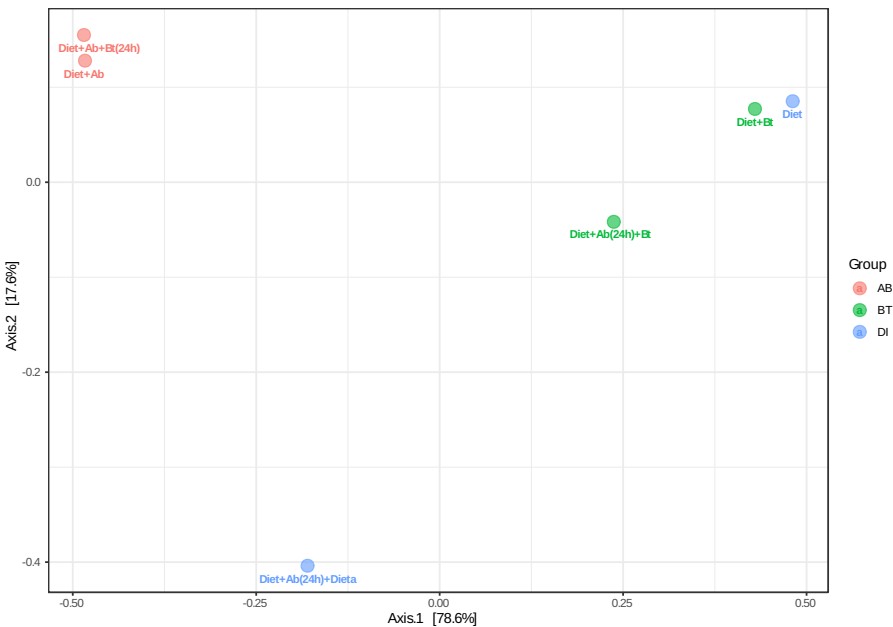

**Figure 7  PCoA analysis of bacterial community made in *S. frugiperda* corn strain.**  The figure shows ordering using Bray distance in 2D [PERMANOVA] $F$-value: 3.0815; R-square: 0.67259; $p$-value $< 0.2$.

## DISCUSSION

When studying the interaction between Bt and gut bacteria, some authors have suggested that gut bacteria is involved in the insect response against this biological control (*Orozco-Flores et al., 2017*; *Paddock et al., 2021*; *Li et al., 2020*). For instance, *Broderick, Raffa & Handelsman (2006)*; *Broderick, Raffa & Handelsman (2006)* found that elimination of gut bacteria in diets supplemented with antibiotics significantly reduces the susceptibility to Bt in *Plutella xyllostela*. Also, in another study, *Li et al. (2021)* demonstrated that *P. xylostella* gut bacteria interact with Bt Cry1Ac protoxin by accelerating their larvae mortality. These

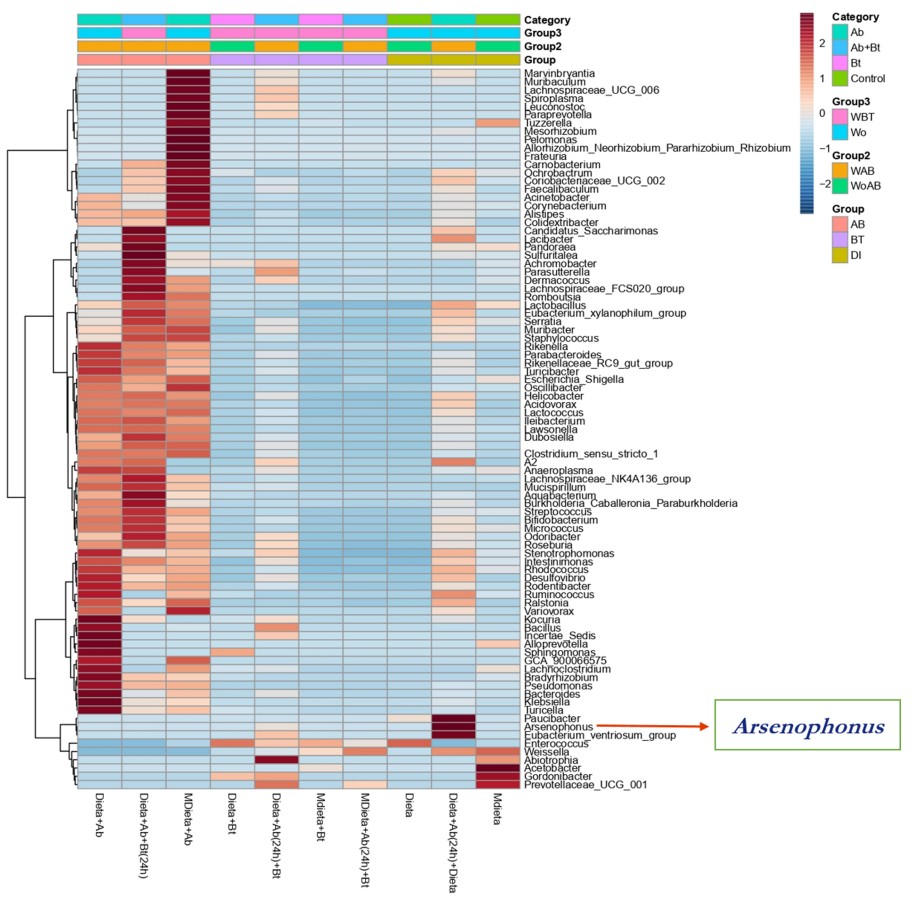

**Figure 8** **Heat map obtained together with ASV assignment obtained for *S. frugiperda* corn strain gut microbiota.** Based on the Euclidean distance metric and the Ward's sum of squares hierarchical clustering method. Distances shown are not phylogenetic but are based on the number of ASV reads within each sample. The scale and legend on the upper right side represent the colors in the heat map associated with the relative abundance of ASVs (clusters of variables on the *Y* axis) within each of the treatments (*X* axis).

**Table 2** **Taxonomic identification of *S. frugiperda* endosymbionts.**

| Isolate ID | Origin | NCBI-GenBank Accession number | Phylogenetic affiliation | % Similarity |
|---|---|---|---|---|
| AMM5A | Male gonads | OP999648 | *Arsenophonus nasoniae* | 100% |
| AFMA | Female gonads | OP999649 | *Arsenophonus nasoniae* | 99.45% |
| AFM3A | Female gonads | OP999650 | *Arsenophonus nasoniae* | 99.86% |
| AFM2A | Female gonads | OP999651 | *Arsenophonus nasoniae* | 99.86% |
| AFM1A | Female gonads | OP999652 | *Arsenophonus nasoniae* | 99.86% |

authors observed that this protoxin causes a dynamic change in *P. xyllostella* midgut and hemocoel microbiota, and also an increment of bacterial load and a reduction in bacterial diversity.

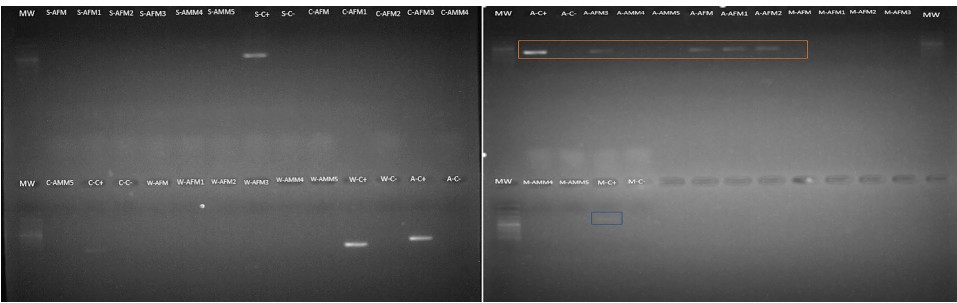

**Figure 9** ***Arsenophonus* detection in *S. frugiperda* corn strain gut microbiota based on specific primers.** Mw: 100 bp ladder. Samples are AFM (adult female gonads), AMM (adult male gonads). The letters that precede the samples refer to the endosymbiont evaluated as follows: S (Spiroplasma), A (Arsenophonus), C (Cardinium), W (Wolbachia) and M (Microsporidia). The amplicon selected in orange correspond to *Arsenophonus*.

On the contrary, other studies have demonstrated that responses (susceptibility/resistance) of insects to Bt were independent of gut microbiota (*Frankenhuyzen, Liu & Tonon, 2010*; *Johnston & Crickmore, 2009*). For this reason, in this study, we followed the methodology conducted by *Orozco-Flores et al. (2017)* in *Plodia interpuctella* as they used six treatments in the presence/absence of Bt and antibiotics to compare the gut microbiota in *S. frugiperda* considering the effect of Bt and antibiotics with and without their interaction. However, in here, no parameters of the immune response were considered as in *Orozco-Flores et al. (2017)*, the relationship between *S. frugiperda* gut bacteria and Bt was analyzed.

Many studies have suggested that the role of gut microbiota in the mode action of Bt proteins is still an ongoing debate. Broderick reports that gut microbiota contributes to the Bt effect in *P. xyllostela* (*Broderick et al., 2009*), meanwhile, other authors have exposed that gut microbiota is not required (*Johnston & Crickmore, 2009*), such as the case of the tobacco hornworm *Manduca sexta*, in which *Raymond et al. (2009)* observed that the gut bacteria are not required for the insecticidal activity of Bt in this pest. In this study, we reported no differences between the microbiota with or without Bt treatment, but we suggest that species of *Enterococcus* must have a role in *S. frugiperda homeostasis* due its permanency on its gut. Further studies are required to carry out Bt bioassays with species of this genus and larvae fitness traits (mortality, longevity, development time, amongst others) in *S. frugiperda*, aditionally studies comparing gut microbiota from both field strain with differing Bt toxin susceptibilities should be considered .

In this work, the identification of corn strain was made in 10 individuals by using a PCR-RFLP of the mitochondrial gene COI and the enzyme *MspI*, suggesting all collected larvae were positive for this marker. Although the number of larvae tested was low, the majority of studies made in Colombia have shown that the corn strain is mainly found in corn crops and the presence of hybrid individuals is low (*Vélez-Arango et al., 2008*; *Cano-Calle, Arango-Isaza & Saldamando-Benjumea, 2015*; *Higuita Palacio et al., 2021*).

The estimation of the LC50 of the commercial product BT-BIOX (Cry1Aa, Cry1Ab, Cry1Ac y Cry2A) was conducted according to concentrations provided by *Ríos-Díez, Siegfried & Saldamando-Benjumea (2012)*. Our results suggest that *S. frugiperda* has developed a rapid tolerance to Bt endotoxins as very high concentrations were needed (20.000 UI/cm3) to produce larvae fitness reduction. This outcome could be explained by the intensive production of corn (variety BT11) and cotton transgenic crops (from 2007 to 2012) in Colombia, at the country regions (Departments) of Cauca (Southeast), Córdoba (Northeast), Meta (West) and Tolima (Center) (*Cuellar Castro, 2015*) and consequently, a high selection pressure was exerted on *S. frugiperda* populations over the last past decade. To evaluate the response to Bt endotoxins in the *S. frugiperda* corn strain, an antibiotic cocktail (gentamicin, streptomycin, penicillin, and rifamycin) used by (*Orozco-Flores et al., 2017*) was employed. However, microbiota eradication was unsuccessful in *S. frugiperda* and two antibiotics were further included in the assay: ampicillin and tetracycline, with a concentration of 5.000 µg/ml each. An antibiogram was used to evaluate the response of gut bacteria to antibiotics and Bt, finding that isolates Sp75 and Sp76 were less susceptible to all Bt concentrations and the cocktail. Additionally, the presence of soil bacteria that are naturally resistant to antibiotics (beta-lactams) such as ampicillin can also play a role in the resistance (*Demanèche et al., 2008*). Isolates Sp75 and Sp76 were identified as *Enterococcus casseliflavus* with a percentage of identity of 90%. *E. casseliflavus* has been also detected: in *Spodoptera litura* larvae (*Thakur et al., 2015*), *Manduca sexta* (Brinkmann et al. 2008), *Spodoptera frugiperda* corn strain (*Higuita Palacio et al., 2021*) and other Lepidoptera species including *Peridroma saucia*, *Bombyx mori*, *Heliothis virescens*, *Hyles euphorbiae*, and *Helicoverpa armigera* (*Mereghetti, Chouaia & Montagna, 2017*). Several species of the genus *Enterococcus* have been reported to be resistant to antibiotics such as cephalosporins, beta-lactams, sulfonamides, and at low levels of aminoglycosides (*Larson et al., 2008*; *Franz et al., 2003*). Future work comparing gut microbiota from both field strain with differing Bt toxin susceptibilities should be considered

According to the results obtained with cultured-dependent methods, the most abundant morphotypes were C1 and C2. Sequencing of the ITS region and 16S genes showed that both corresponded to *E. mundtii* and *E. casseliflavus* respectively. *E. mundtii* was detected in treatments Diet + Bt and Diet + Ab (24h) + Bt suggesting that this species tolerates Bt endotoxins and antibiotics. *Higuita Palacio et al. (2021)* also found that *Enterococcus* was the most abundant genus in the *S. frugiperda* corn strain from Colombia, including the species *E. mundtii* and *E. casseliflavus*. *Higuita Palacio et al. (2021)* used MacConkey selective media to grow Gram-negative bacteria in Petri dishes. They also used nutritive agar as non-selective media and identified other genera such as *Klebsiella*, *Enterobacter*, and *Bacillus*. In this study, only LB media was used for colony identification, and for this reason, only Gram-positive and cocci bacteria were found (*Enterococcus*). They sequenced the V4 region of the 16S ribosomal gene through NGS sequencing and found that the most abundant genera were *Enterococcus* and *Erysipelatoclistridium*. In the specimens and assays reported here, the genus *Erysipelatoclistridium* did not show higher predominance but the genera *Enterococcus*, *Klebsiella* (Enterobacteriales: Enterobacteriaceae), and Enterobacter

(Enterobacterales: Enterobacteriaceae) were consistently detected as abundant microbiome members.

Moreover, *Almeida et al. (2017)* isolated and assessed the pesticide-degrading capacity of gut bacteria from *S. frugiperda* larvae under laboratory conditions. They identified *E. mundtii* y *E. casseliflavus* in populations of this insect collected in corn crops from Brazil. They tested these two species against lambda-cyhalothrin, deltamethrin, chlorpyrifos ethyl, spinosad, and lufenuron to determine whether they can tolerate and degrade these pesticides and also reported the species: *Microbacterium paraoxydans*, *Delftialacustris*, *Leclercia adecarboxylata*, *Pseudomonas stutzeri*, *Arthrobacter nicotinovorans*, *Pseudomonas psychrotolerans*, *Staphylococcus sciuri* subspecies *sciuri* and *Microbactetium arborescens* in FAW. These authors concluded that *S. frugiperda*-resistant strains represent an excellent reservoir of insecticide-degrading bacteria with bioremediation potential. In another study, *Oliveira (2021)* reported *Enterococcus* and *Pseudomonas* from *S. frugiperda* collected in corn fields from Brazil, Colombia, Mexico, Panamá, Paraguay, and Perú. According to them, in Lepidoptera, the gut microbiota is characterized as having low diversity indexes and this could be due to their gut being relatively short and the pH being mainly basic.

*E. mundtii* has also been isolated in *Spodoptera littoralis*. This species is a strong competitor against other gut bacteria (*e.g.*, antimicrobial activity) (*Shao et al., 2017*). *E. mundtii* cells accumulate on the epithelial gut forming a biofilm on the host gut (*Shao et al., 2014*). Also, other reports have shown that *E. mundtii* secretes a bacteriocin that inhibits the colonization of other pathogens impeding the presence of *B. thuringiensis* in insect guts (*Li et al., 2020*). *E. casseliflavus* was also detected in treatments: Diet, Diet + Ab (24h), and Diet + Ab (24h) + Bt demonstrating less susceptibility to Bt endotoxins. *Paddock et al. (2021)* have suggested that microbial gut communities remain almost the same between bioassays with and without Bt endotoxins suggesting many of them are resistant to this biological control, this is the case of *E. mundtii* and *E. casseliflavus* in *S. frugiperda*.

The third morphotype found was C3 (the less abundant) and it was genetically identified as *E. silesiacus* with a 99.6% of identity in the treatment Diet + Bt. Nonetheless, this morphotype was also identified as *E. mundtii* with a 99.3% of genetic identity in treatments Diet + Ab (24h) and Diet + Ab (24h) + Bt. This result could be due to the quality of the sequences (sizes from 600 to 900 bp) and to the high genetic similarities amongst species of the genus *Enterococcus*. Sequencing of the 16S rRNA gene is one of the most commonly used methods to identify bacteria and it is also employed for phylogenetic analyses (*Woese, 1987*). However, many related species share high genetic similarities making it difficult to identify and differentiate them (*Wang et al., 2007*). To solve this problem, sequencing of the gyrase B (gyrB) gene was also performed in this work, allowing us to identify one isolate from these morphotypes (Table 1).

In a previous investigation made on *S. frugiperda* gut microbiota identification by *Higuita Palacio et al. (2021)* conducted in Colombia, they used OTUS to identify microbiota by using NGS sequencing metadata. Here, ASVs (amplicon sequencing variants) were used instead. Given that differences in detection methods were used between these two studies, only in this work, we reported for the first time the genus *Arsenophonus*. However, specific primers were also used in this study corroborating that this endosymbiont is present in FAW

microbiota. Here we reported a conserved microbial core under controlled conditions, similar results were obtained by *Higuita Palacio et al. (2021)* for the microbial core as they also studied the gut microbiota in *S. frugiperda* corn strain in larvae populations analyzed directly from corn fields in Antioquia department (Northern Colombia).

Alpha diversity results showed that antibiotics assays exhibited the highest diversity values, suggesting that the cocktail drastically reduced the genus *Enterococcus* populations, the most abundant genus found in *S. frugiperda* gut microbiota. By removing *Enterococcus*, colonization of new gut bacteria was likely to occur in this insect, suggesting strong interspecific competition of gut bacteria in this pest, particularly by this genus. According to *Orozco-Flores et al. (2017)*, some studies (*Broderick, Raffa & Handelsman, 2006*) have suggested that in Lepidoptera, gut bacteria are important in the response to Bt but *Johnston & Crickmore* in *2009*, have found that the susceptibility of insects to Bt were independent of microbiota and associated with the effect of antibiotics on the insect. Therefore, six treatments were used in this work to be able to compare gut microbiota detected in the presence/absence of Bt and antibiotics. *Paddock et al. (2021)* observed that resistance to Bt produces alterations in the microbiome of *Diabrotica virgifera* (the western corn rootworm) that may contribute to resistance to this biological control and also that susceptible insects harbored more microbiota compared to resistant ones, thus our results validate their work since the bacterial community composition decreased in resistant *S. frugiperda* larvae to Bt, suggesting that resistant *Enterococcus* species predominated as the most adapted group to selection pressure.

Results on beta diversity estimations showed no significant differences among the treatment groups. However, when comparisons were conducted between samples in the presence/absence of antibiotics, diversity estimations were significantly different because, in diets free of antibiotics, diversity was lower in comparison to antibiotics-supplied diets, particularly those supplemented with Bt. This result could be due to the antibiotic cocktail used in this work eradicating the most abundant and competitive bacteria from the *S. frugiperda* gut, which were gram-positive. In a recent study made with *P. xyllostella*, *Li et al. (2021)* showed that Bt Cry1Ac protoxin interacts with the gut microbiota and accelerates the mortality of *P. xylostella* larvae. Cry1Ac protoxin was found to cause a dynamic change in the midgut and hemocoel microbiota of *P. xylostella*, with a significant increase in bacterial load and a significant reduction in bacterial diversity. In turn, loss of gut microbiota significantly decreased the Bt susceptibility of *P. xylostella* larvae. *Li et al. (2021)* also introduced three gut bacterial isolates *Enterococcus mundtii* (PxG1), *Carnobacterium maltaromaticum* (PxCG2), and *Acinetobacter guillouiae* (PxCG3) restored sensitivity to Bt Cry1Ac protoxin. Despite their results, a realistic application in the field can be difficult and requires more studies.

In *Spodoptera litoralis* the most abundant genera are *Enterococcus*, *Klebsiella*, *Enterobacter*, and *Pseudomonas* (*Chen et al., 2016*). The results obtained in this work mostly showed the predominance of the genus *Enterococcus* in *S. frugiperda*. *Mereghetti, Chouaia & Montagna (2017)* demonstrated that in Lepidoptera, the gut microbiota has relatively low diversity. In their study, they included the species *Spodoptera littoralis*, *Helicoverpa armigera*, *Heliothis virescens*, *Busseola fusca*, *Agrotis ípsilon*, *Ostrinia nubilalis*, *Plutella*

*xyllostella,* and *Manduca sexta* and they identified the families Enterobacteriaceae and Enterococcaceae, followed by Bacillaceae (except in *H. virescencens*) and Micrococcaceae (except in *H. virescence, O. nubilalis,* and *P. xylostella*). According to them, gut bacteria communities in Lepidoptera are influenced by their feeding behavior, diet, developmental stage, and origin (laboratory colony, field populations). Forest species such as *Lymantra dispar, Brithy scrini, Choristoneura fumiferana,* and *Thaumatopoea pityocampa* exhibited different microbiota composition, depending on their polyphagia as the more hosts they consumed, the more diverse microbiota they harbor. Likewise, in pests of economic importance such as *O. nubilalis, P. xylostella, M. sexta, H. armigera, S. littoralis, B. fusca, H. virescens,* and *A. ipsilon* microbiota composition also differed between polyphagous and oligophagous species *Mereghetti, Chouaia & Montagna (2017).*

Finally, in this study endosymbiont detection was possible for the genus *Arsenophonus* by using both NGS and specific primers. This endosymbiont might play an important role in the response of *S. frugiperda* to insecticides and Bt as found in other insects (*Tyagi et al., 2022*). *Wolbachia* detection in *S. frugiperda* collected from diverse locations has been successful when using genomic DNA from abdominal samples and sequencing of PCR amplicons (*Schlum et al., 2021*), *S. litura* and *S. exempta* by using Multilocus Sequence Typing (MLST) methodologies (*Salunkhe et al., 20l4; Graham et al., 2012*). Nevertheless, *Dumas et al. (2015)* did not detect this bacterium in *S. frugiperda* by using NGS methods. Differences in the results obtained might be due to the origin of these insects as the first two species originated from the Eastern hemisphere and the last one from the Western hemisphere. According to *Qu et al. (2013), Werren, Skinner & Huger (1986),* and *Huger, Skinner & Werren (1985)* the presence of *Arsenophonus nasoniae* is correlated to 80% of male mortality in *Nasonia* sp., suggesting that this endosymbiont is relevant in this wasp. Detection of this endosymbiont is important for *S. frugiperda* studies based on biological control as endosymbionts have been previously used to affect insect density populations in nature.

## CONCLUSIONS

This study demonstrated the presence of two species of *Enterococcus* in *S. frugiperda* gut microbiota that are capable of surviving both Bt and antibiotics, these are *E. mundtii* and *E. casseliflavus*. Also, the endosymbiont *Arsenophonus* was detected in this pest. These bacteria were identified by using culture-dependent and culture-independent methods. Further studies are necessary to determine if *Wolbachia* or other endosymbionts are also present in this moth and if manipulation of microbiota composition can be employed for the improvement of its management in nature.

## ACKNOWLEDGEMENTS

The authors would like to thank the Laboratorio de Biología Celular y Molecular at Universidad Nacional de Colombia, Sede Medellin where molecular biology experiments were performed, and the laboratories Procesos Moleculares and Ecología y Evolución de insectos where bioassays were made.

### Funding

This work was funded by MinCiencias (2018-2023) under the project entitled: Bioprospección de la Microbiota Asociada a Insectos Plaga De Cultivos De Interés Agrícola en Colombia: Spodoptera Frugiperda (Biotipos Maíz Y Arroz) Y Trips Del Aguacate Para El Desarrollo De Alternativas De Manejo De Su Control (80740-146-2019). Decision to publish and sample collection for data

### Grant Disclosures

The following grant information was disclosed by the authors:
MinCiencias (2018–2023).
*Spodoptera Frugiperda* (biotipos maíz y arroz) y Trips Del Aguacate Para El Desarrollo De Alternativas De Manejo De Su Control:  80740-146-2019.

### Competing Interests

The authors declare there are no competing interests. Howard Junca is employed by Microbiomas Foundation.

### Author Contributions

- Yuliana Castañeda-Molina performed the experiments, analyzed the data, prepared figures and/or tables, authored or reviewed drafts of the article, and approved the final draft.
- Sandra María Marulanda-Moreno performed the experiments, authored or reviewed drafts of the article, and approved the final draft.
- Clara Saldamando-Benjumea conceived and designed the experiments, analyzed the data, prepared figures and/or tables, authored or reviewed drafts of the article, and approved the final draft.
- Howard Junca analyzed the data, prepared figures and/or tables, authored or reviewed drafts of the article, and approved the final draft.
- Claudia Ximena Moreno-Herrera conceived and designed the experiments, authored or reviewed drafts of the article, and approved the final draft.
- Gloria Cadavid-Restrepo conceived and designed the experiments, analyzed the data, authored or reviewed drafts of the article, and approved the final draft.

### Field Study Permissions

The following information was supplied relating to field study approvals (Permiso marco de recolección de especímenes silvestres, resolución 0255, 14/03/2014 (art.3)):

The collection of the larvae and genetic access was provided by ANLA (Autoridad Nacional De Licencias Ambientales to Universidad Nacional de Colombia.

### DNA Deposition

The following information was supplied regarding the deposition of DNA sequences:

All bacteria sequences are available at GenBank OP999646–OP999647, OQ344654–OQ344662 and OQ351359.

Arsenophonus in Fall armyworm sequences are available at OP999648–OP999652.

## Data Availability

The raw data are available in the Supplementary File.

The paired-end reads obtained from amplicon sequencing for each sample (Raw data), which were used for the processing and analysis conducted using DADA2 (https://www.nature.com/articles/ismej2017119) are available in the Supplementary Files.

The pipeline described in the tutorial (https://benjjneb.github.io/dada2/tutorial.html) was followed, with the modification of using the latest versions of the taxonomic references, RDP 18 (https://sourceforge.net/p/rdp-classifier/news/2020/07/rdp-classifier-213-july-2020-release-note/) and Silva 138 (https://www.arb-silva.de/documentation/release-1381/) available at Zenodo: McLaren, Michael R., & Callahan, Benjamin J. (2021). Silva 138.1 prokaryotic SSU taxonomic training data formatted for DADA2 [Data set]. Zenodo. https://doi.org/10.5281/zenodo.4587955.

## Supplemental Information

Supplemental information for this article can be found online at http://dx.doi.org/10.7717/peerj.15916#supplemental-information.

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
