# Peer review of "Microbiome analysis of Spodoptera frugiperda (Lepidoptera, Noctuidae) larvae exposed to Bacillus thuringiensis (Bt) endotoxins"

_PeerJ, doi:10.7717/peerj.15916_

## Round 0.1 · original submission · Major Revisions

The reviewers (experts in the field) have made a number of suggestions to improve the manuscript so that it is suitable for publication. Please consider and address each comment carefully.

Reviewer 1 ·

Basic reporting

• Overall, the manuscript is well written and understandable
• Line 1: Different title listed than what was submitted
• Background: only the last sentence mentions gut microbiota, but this is the focus of the manuscript. It would be good to use this section to portray the importance of studying gut microbiota in this pest (i.e. why and how “Gut microbiota can be potentially used to improve integrated pest management”)
• Lines 327-363: Discussion of Fig. 2 and 3 having multiple parts, but there is only 1 graph in Fig. 2 and 3?
• Lines 378-388: This paragraph is better suited for the Introduction section.

Experimental design

• No larvae directly from the field were preserved and the gut microbiota sequenced? Sequencing appears to have been performed after 24h of treatments, but the gut residence time is less than this; how would such a long incubation time in a lab setting have affected the gut microbiota community compared with field S. frugiperda?

Validity of the findings

• The authors’ main hypothesis appears to be that the control of the gut microbiota can be used to aid S. frugiperda pest management (presumably using Bt toxins, which are tested therein). However, another study (www.pnas.org/cgi/doi/10.1073/pnas.1707186114) and several studies noted in the manuscript suggest gut microbiota in lepidopterans is transient and plays an independent role in the survival of these insects. There should be some discussion on these conflicting theories particularly since the authors do not provide strong literature that demonstrates how manipulating gut microbiota can improve S. frugiperda control using Bt toxins. For example, the Li et al. (2021) study only demonstrates that without a gut microbiota community Cry1Ac is less effective; how can you make Cry1Ac more effective?
• Line 535: This sentence is unclear: “…manipulation of gut microbiota can be used as an alternative…” refers to Cry1Ac? If so, the authors need to clarify how manipulating gut microbiota without Cry1Ac can manage S. frugiperda?
• Line 578-9: How can the bacteria identified in this study be “used for biological control of this insect”?

Reviewer 2 ·

Basic reporting

The manuscript reports the identification of bacteria isolated from the digestive tract of fall armyworm (FAW) larvae from Colombia. Both culture-dependent and independent methods were used in identifying microbes after treatment with/without antibiotics and/or exposure to a Bt pesticide. This information may be publishable and useful in helping describe the FAW microbiome, but authors make some claims on its utility for FAW control and effects on Bt susceptibility that are not supported by experimental evidence. English usage was acceptable, although the text should be carefully revised to improve clarity (see examples below) and unfinished sentences (for instance lines 110-111).
The title is unclear and misleading. The authors did not find any significant effect associated to treatment with the Bt pesticide, yet they refer to this treatment in the title as inducing “low modulation”?. Also in the same title Bt is combined with presence of Arsenophonus and Enteroccoccus resistance to antibiotics, which are not related. A more accurate and clear description of the work would be: “Detection of Arsenophonus symbiont in gonads and of antibiotic-resistant Enteroccoccus in the gut microbiome of Spodopera frugiperda”.
The Hofte & Wihteley 1989 reference is too old and should be replaced by a more recent review of Bt activity, for example Palma et al 2014, Toxins.
The authors claim that Wolbachia has not been reported in FAW, but there is a report that shows detection of Wolbachia in field-collected FAW samples (Schlum et al, BMC Genomics 2021). A citation should be included and the text corrected accordingly. Labeling in Figure 9 should be improved for clarity: if the names on the gel lanes are not important they could be replaced by numbers or letters. If the names should be retained a letter could be used and the full name could be detailed in the figure legend.
The third determinant in the Cry toxin nomenclature should not be capitalized. Thus, throughout the text Cry1AC and Cry1AB should be corrected to Cry1Ac and Cry1Ab, respectively.
Throughout the text the authors use “resistance” where “less susceptible” or “less sensitive” should be used. For instance, line 424 states “Sp75 and Sp76 were resistant to all Bt concentrations”. There is no evidence that these bacteria were ever susceptible to Bt, so using the word “resistance” here is not appropriate. The same applies to lines 72 (cron versus rice susceptibility, not resistance) and 478. Also in line 576 change “resist” to “survive”.
Line 56-57: “Polyphagous” and “feed from several crops” is redundant. This is again repeated in the next sentence (can feed ON over 350 plant species). Please revise the text to avoid redundancy.
Line 60: To this reviewer’s knowledge, the origin of invasive FAW has not been elucidated and based on trade routes probably involved FAW from South America, consequently the assertion used (from contaminated material from the US) should be deleted.
Line 82: Most Bt crops control above ground pests, please correct.
Line 115-116: This sentence is confusing. Bt crops are active against FAW, what do the authors mean stating that FAW have not been affected by Bt crops in Colombia?
Lines 229-232: This paragraph should be combined with the section where this DNA was used for an analysis.
Line 260: How many individuals used?
Lines 378-379: This sentence is confusing and inaccurate. The widespread use of Bt crops has not generated resistance to non-target insects. FAW is a target for some Bt corn varieties (depending on toxins expressed).
Line 512: Please cite these investigations.

Experimental design

The study seems rigorous in its design and analysis, but there are a couple of concerns with the experimental design. There is no description of how the different treatments affected the FAW larvae. This information is needed to estimate if the presence/absence of specific bacteria influence FAW. Without this information, claims in lines 569-571 are not supported (how would Enterococcus be used to control FAW if it is a symbiont, and its absence does not have an effect on FAW?). Also, in Line 186: why were plates incubated at 37C? This is not a biologically-relevant temperature from insect bacteria and probably eliminated some of the bacteria from being cultured.
It seems originally designed to detect alterations in the FAW microbiome produced by exposure to Bt toxins and antibiotics, but results only detected relevant effects of antibiotics, which would be predicted. Thus, the research question considered is unclear and the study appears exploratory in nature rather than hypothesis-driven.
Some relevant details are missing from the methods section. For instance, the generation of the FAW strain used is confusing. Insects were collected from two locations and then pooled into a single colony in the lab? How many generations was it maintained in the lab before analysis?
Line 154: how was the LC50 dose estimated? Authors need to consider that they used 3rd instar larvae, which are significantly less susceptible to Bt, explaining their observation of low susceptibility and not supporting claims of resistance (stated in line 414).

Validity of the findings

Replications, controls and statistical analyses are sound.
Some of the conclusions and claims are confusing or not supported by experimental evidence. In line 292 authors state antibiotics eliminated the microbiota, but in line 358 they state highest diversity was observed with antibiotic treatments. This is confusing, which was true?
Line 368: What do the authors mean by “bioprospection potential”? What is the evidence supporting that this bacterium has this potential?
Line 426: This is inaccurate. What do the authors mean by the bacteria adapting to transgenic crops? These crops produce Bt toxins, which may or may not be toxic to different bacteria, but this does not mean that bacteria not being affected have become “resistant” due to Bt crops. Furthermore, in lines 428-429 the authors claim that resistance genes in Bt crops may explain resistance in the bacterial isolates. This seems difficult to support, how do these genes become adopted by the bacteria?
Line 482: There is no evidence in the literature for the claim that resistance in Bt favors toxin-degrading bacteria, the citation used (Vasquez et al 2012) is not related to Bt.
Lines 569-571 and 579: It is unclear how these bacteria could be used for FAW control since they are symbionts in its gut? There is no evidence that their elimination/addition has an effect on the insect.

---

## Round 0.2 · Minor Revisions

Some additional changes are recommended, please see the reviewer comments. In addition, I do not think that the new title is better, you may consider just the first part of the title "Microbiome analysis of Spodoptera frugiperda (Lepidoptera, Noctuidae) larvae exposed to Bacillus thuringiensis (Bt) endotoxins".

Reviewer 1 ·

Basic reporting

Overall, the manuscript is well written and understandable

Experimental design

• Only two populations of FAW were sampled and only from corn fields. Given the contrasting insecticidal tolerances among FAW populations noted by the authors how do the authors anticipate the gut microbiota to differ in other FAW? I suggest the authors consider for future work comparing the gut microbiota of two FAW populations from the field with differing Bt toxin susceptibilities (e.g. the corn- vs. rice-adapted FAW populations).

Validity of the findings

• The authors have improved the manuscript by including additional supporting studies for the importance of studying gut microbiota, but I remain skeptical of the potential to use endosymbionts in countering Bt toxin resistance (or for biological control of lepidopterans in general) and do not believe the authors have supported how this can be achieved. In contrast the authors have correctly included conflicting literature demonstrating endosymbionts may or may not influence biocontrols such as Bt toxins. Therefore, the following clarifications should be made:
o Lines 51-3: delete “… demonstrating the relevance of the results obtained in this work.” Previous studies may suggest a theory, but this is not demonstrated relevance.
o Line 118: modify “…promising new biocontrol approach FOR THE SPREAD OF PATHOGEN DISEASE..”
o Line 141: modify “Gut microbiota manipulation MAY be a promising method…”
o Line 391: delete the sentence “This result represents the first report of an endosymbiont with biocontrol potential…”
o Line 519: delete “…with biocontrol potential.”
o Line 588-9: modify “…endosymbionts have been previously used to AFFECT insect density populations in nature.” The only evidence provided for this is in the case of Wolbachia and mosquitoes. While technically true, this appears to have more to do with limiting pathogen disease; mosquitoes populations lifespans are decreased, but not enough to result in any meaningful insect “control”. In addition this may not be relevant to a lepidopteran pest such as FAW.
o Lines 596-7: delete “…suggesting they can be potentially used for biological control of this insect.

• In lines 551-557 the authors’ discussion of the results from Li et al 2021 is misleading by suggesting the introduction of bacteria can be used to help control P. xylostella. Their results restoring sensitivity to Cry1Ac by introducing bacterial isolates was after antibiotic treatment of the larvae, and, thus has no relevance to a realistic biological control application (unless the authors are suggesting using widespread antibiotics to sterilize lepidopterans in the field?).

• Lines 596-7: “…suggesting they can be potentially used for biological control of this insect.” This should be removed since the authors have not demonstrated how this can be achieved.

Additional comments

o In order to demonstrate “new strategies for biological control” using endosymbionts it might help to find or generate a study that demonstrated Bt toxin resistance was linked to the gut microbiota. Most Cry toxin resistance is linked to decreased binding in the gut epithelium – often due to receptor modification, but perhaps endosymbionts could play a role here as well?

o The authors seem to be focused on bacterial endosymbionts, but what about viruses or fungi? Baculovirus, in particular, are very relevant for lepidopteran control.

---

## Round 0.3 · accepted · Accept

Thank you for carefully revising according to the reviewer comments. The manuscript is now suitable for publication in PeerJ.